

# Academic stress in college students: descriptive analyses and scoring of the SISCO-II inventory

Juan-Luis Castillo-Navarrete[1,2,3], Claudio Bustos[1,2,4], Alejandra Guzman-Castillo[1,2,5] and Walter Zavala[6]

[1] Programa Neurociencias, Psiquiatría y Salud Mental, NEPSAM, Universidad de Concepción, Concepción, Chile
[2] Programa Doctorado Salud Mental, Departamento de Psiquiatría y Salud Mental, Facultad de Medicina, Universidad de Concepción, Concepción, Chile
[3] Departamento de Tecnología Médica, Facultad de Medicina, Universidad de Concepción, Concepción, Chile
[4] Facultad de Ciencias Sociales, Universidad de Concepción, Concepción, Chile
[5] Departamento de Ciencias Básicas y Morfología, Facultad de Medicina, Universidad Católica de la Santísima Concepción, Concepción, Chile
[6] Carrera de Fonoaudiología, Facultad de Ciencias de Salud y Ciencias Sociales, Universidad de Las Americas, Concepción, Chile

Corresponding authors
Juan-Luis Castillo-Navarrete, jucastillo@udec.cl
Claudio Bustos, clbustos@udec.cl

## ABSTRACT

In a competitive and demanding world, academic stress is of increasing concern to students. This systemic, adaptive, and psychological process is composed of stressful stimuli, imbalance symptoms, and coping strategies. The SISCO-II Academic Stress Inventory (SISCO-II-AS) is a psychometric instrument validated in Chile. It evaluates stressors, symptoms, and coping, both individually and globally. For its practical interpretation, a scale is required. Therefore, this study aims to descriptively analyze the SISCO-II-AS and to obtain its corresponding scales. Employing a non-experimental quantitative approach, we administered the SISCO-II-AS to 1,049 second and third-year students from three Chilean universities, with a disproportionate gender representation of 75.21% female to 24.79% male participants. Through descriptive and bivariate analysis, we established norms based on percentiles. For the complete instrument and its subscales, significant differences by sex were identified, with magnitudes varying from small to moderate. For the full instrument and its subscales, bar scale norms by percentile and sex are presented. Each subscale (stressors, physical and psychological reactions, social behavioural reactions, total reaction, and coping) has score ranges defined for low, medium, and high levels. These ranges vary according to the sex of the respondent, with notable differences in stressors and physical, psychological, and social behavioural reactions. This study stands out for its broad and heterogeneous sample, which enriches the representativeness of the data. It offers a comprehensive view of academic stress in college students, identifying distinctive factors and highlighting the importance of gender-sensitive approaches. Its findings contribute to understanding and guide future interventions. By offering a descriptive analysis of the SISCO-II-AS inventory and establishing bar norms, this research aids health professionals and educators in better assessing and addressing academic stress in the student population.

# INTRODUCTION

Mental health takes on special relevance in the case of college students. Currently, a growing number of young people are entering higher education seeking academic training and access to better job opportunities. However, this process can be affected by various factors that can trigger mental health problems, negatively impacting performance and retention at university. College students often face multiple demands in terms of academic load, work responsibilities, social life, and family, which can result in increased academic stress (AS) (*Barraza-Macías, 2007a*; *Putwain, 2007*). AS gradually increases due to increasing academic demands, pressure for grades, competition, and fear of not meeting expectations, affecting student performance and health (*Castillo-Navarrete et al., 2020*). Consequently, it has been observed that students facing high levels of AS are more likely to experience anxiety, depression, insomnia, and other mental disorders. These conditions can hurt both a student's academic performance and general well-being, underscoring the critical need to comprehensively address AS in the educational environment (*Castillo-Navarrete et al., 2023a*).

The conceptual grasp of 'academic stress' is complicated not only by the term's broad use but also by the diversity of evaluation tools employed. Rather than relying on a single uniform method, researchers utilize a range of instruments such as the Academic Stress Inventory (ASI), Beck Anxiety Inventory (BAI), and the Perceived Stress Scale (PSS), to name a few (*Cohen, Kamarck & Mermelstein, 1983*; *Beck et al., 1988*; *Polo, Hernández López & Pozo Muñoz, 1996*). Each tool brings its lens to the phenomenon, measuring varying dimensions from cognitive and emotional responses to physiological effects, thus contributing to inconsistent outcomes and complicating cross-study comparisons (*Barraza-Macías, 2007a*; *Barraza-Macías, 2007b*).

The term 'academic stress' is often used broadly, yet its actual scope and limitations may remain elusive. This ambiguity is exacerbated by the variety of terms used to describe the concept, such as 'student,' 'university,' 'burnout,' 'school,' and 'exam stress,' leading to an unclear conceptualization and an emphasis on stressors and symptoms rather than on a holistic understanding. This lack of a unified definition is acknowledged as a significant obstacle in the literature. For instance, *Liu, Ji & Zhang (2023)* investigates how general self-efficacy and depression—shaped by factors including gender—can substantially influence academic stress. Such studies underscore the necessity for a more comprehensive and nuanced understanding of this complex phenomenon.

Therefore, a multidimensional approach is needed to understand and address AS. However, the measurement of stress in college students has been based on tools that provide simple measurements with little contextualization. There are also tools to assess stressful situations related to academic, family, and economic aspects. Some examples of these are the Student Life Stress Inventory (*Gadzella, 1994*), Undergraduate Stress

Sources Questionnaire (*Blackmore, Tucker & Jones, 2005*), Academic Expectations and Stress Inventory (*Ang & Huan, 2006*) and College Student Stress Scale (*Feldt, 2008*). In contrast, others have focused on the stressful potential of different academic conditions, such as the Academic Stress Scale of the Academic Stress Questionnaire *Cabanach et al. (2010)*; *González Cabanach et al. (2010)*; *Cabanach, Souto-Gestal & Franco (2016)*.

In this context, *Barraza-Macías (2007a)* proposed a more comprehensive and processual theoretical model of AS. He defines AS as a systemic, adaptive, and psychological process. This process consists of three moments: (i) the confrontation with demands perceived as stressors; (ii) a systemic imbalance that manifests itself in the form of symptoms; and (iii) a response aimed at re-establishing the balance. Thus, three components are identified in the systemic process: stressful stimuli, symptoms indicating imbalance and coping strategies (*Barraza-Macías, 2007b*).

Consequently, Barraza-Macías developed the SISCO inventory of academic stress (SISCO-AS). This self-descriptive psychometric instrument is based on a three-factor structure: stressors, symptomatology, and coping (*Barraza-Macías, 2007b*). In Chile, the SISCO-AS has been used and its psychometric properties have been studied (*Guzmán-Castillo et al., 2018*; *Guzmán-Castillo et al., 2022*). In 2020, a modification of the SISCO-AS, the SISCO-II inventory of academic stress (SISCO-II-AS), was introduced. This new version maintains the subscales of stressors and coping. In addition, it identifies two factors in the symptomatology subscale (now called Total Reaction): physical and psychological reactions, and social behavioral reactions (*Castillo-Navarrete et al., 2020*).

Assessing the level of AS in diverse student populations requires appropriate scales. When studying the psychometric properties of an instrument, several aspects should be considered. These include item analysis, estimating the reliability of the scores, and obtaining evidence of validity (*Muñiz & Fonseca-Pedrero, 2019*). This last aspect involves the study of dimensionality, the analysis of the differential functioning of the items and, the relationship with external variables. In essence, it refers to the quality of the inferences made from the scores (*Muñiz & Fonseca-Pedrero, 2019*; *Prieto Adánez & Delgado González, 2010*). In the case of the SISCO-II-AS, it is necessary to execute the instrument's baremization. This allows for establishing the necessary cut-off points to interpret the scores obtained, facilitating its practical use (*Muñiz & Fonseca-Pedrero, 2019*).

Having adequate scales is essential for any instrument intended for a specific study population. The direct score obtained by an individual is not directly interpretable. Therefore, it is necessary to refer it to other individuals within the same normative group. Scales provide information on the position of an individual about the rest of the normative group, assigning a numerical value to each possible score. There are several ways to obtain scales for an instrument, including chronological (age) scales, percentiles, and typical (standardized and/or normalized) scores (*Muñiz & Fonseca-Pedrero, 2019*; *Prieto Adánez & Delgado González, 2010*).

In this context, to evaluate diverse student populations about their level of AS, it is necessary to have adequate scales. Therefore, this paper aims to descriptively analyze the SISCO-II-AS and obtain its corresponding scales, based on the sample used when this inventory was reported (*Castillo-Navarrete et al., 2020*). Following this theoretical

framework, the paper details the methods of data collection, participant demographics, and the statistical analysis process utilized. Subsequently, the study results are presented, which address the efficacy of the SISCO-II-AS inventory and establish the corresponding normative scales. The discussion then elucidates the implications of these findings within the current academic milieu, summarizing the study's pivotal points and proposing avenues for future research.

## METHODOLOGY

### Participants

This study, quantitative in nature and non-experimental design, was conducted with a purified sample of 1,049 students. The participants were second and third-year students from three Chilean universities: Universidad de Concepción (UdeC), Católica de la Santísima Concepción (UCSC), and del Desarrollo (UDD). Each student expressed their willingness to participate by signing an informed consent form and a general datasheet. This form was approved by the Scientific Ethical Committee of the Faculty of Medicine of the Universidad de Concepción (No CE 65/2018). Individuals who declared to be under psychological and pharmacological treatment as an important resource to manage their mental health were included. These individuals are part of the student reality and can perform adequately from an academic point of view (*Fu & Qiao, 2023*; *Lieslehto et al., 2023*; *Tang et al., 2023*). Excluding them from the sample could bias the results concerning the university student population as a whole. The gender composition of our sample is representative of the student populations within the faculties and programs to which we had access. A predominance of female participants is consistent with demographic trends observed in similar academic settings, as reported in our previous studies (*Castillo-Navarrete et al., 2023a*). This demographic characteristic is an important consideration for interpreting the study's findings within the broader context of academic stress research.

### Instrument

The SISCO-II-AS consists of 33 items. The first item, dichotomous (yes-no), determines whether the respondent continues to answer. The second item identifies the overall self-perception of the level of AS. Eight additional items seek to identify the frequency of environmental demands perceived as stressors. Another 17 items determine the frequency of symptoms or responses to stressful stimuli. Finally, six items seek to identify the frequency of use of coping strategies. The last three sections use a Likert scale (1: never, 5: always). The parts of this instrument can be used in isolation, combined or as a whole (*Castillo-Navarrete et al., 2020*).

### Procedure

The research team applied this instrument during the student's second or third year of study. The careers included were Medical Technology, Obstetrics and Childcare, Kinesiology, Phono audiology, Nursing, Dentistry, Chemistry and Pharmacy, and Nutrition and Dietetics. The application of the instrument lasted approximately 10 min and was carried out at the end of 2018.

The data used are available at https://doi.org/10.48665/udec/M6681K

## Statistical analysis

Descriptive analysis was performed using measures of central tendency and dispersion. Categorical variables were analyzed using frequency and percentage. Bivariate and multivariate analyses were performed, comparing groups using each variable separately. Bar norms were established based on percentiles, both for each part of the SISCO-II-AS, individually, combined and as a complete instrument. A significance level of $\alpha = 0.05$ was established. Data were coded in Microsoft Excel and statistical analysis was run with R Studio (*R-Project, 2023*).

## Handling of missing data

A total of 38 cases had missing data. Specifically, 36 of them had only one missing value, and the remaining two cases had two missing items. Multiple Imputation by Fully Conditional Specification was employed, using the 'mice' package in R, with 21 imputed datasets and 25 iterations, ensuring the convergence of the imputation chains. To combine the results of the multiple imputations, Rubin's methodology was used (*Rubin, 1987*).

## RESULTS

The sample presented a gender distribution of 75.21% women (789) and 24.79% men (260) (Table 1). According to the university of origin, 49.38% (518) belonged to UdeC, 12.58% (132) to UCSC and 38.04% (399) to UDD. The average age of the students was 21.26 years (SD =1.81), with a range of 18 to 34 years. The majority were in their third to sixth semester, while a small group (33 students) were in their seventh semester or higher.

When evaluating SISCO-II-AS scores by sex, significant differences ($d = 0.56$) were found on the full instrument and subscales. Specifically, the differences are relevant to stressors, physical and physiological reactions, and social behaviour (Table 2). It is important to note that these differences vary in magnitude. They are small for social behavioural reactions ($d = 0.237$) and moderate for stressors ($d = 0.468$) and physical and psychological reactions ($d = 0.631$). It is relevant to consider the moderate size difference in the total reaction ($d = 0.518$). This masks gender differences in both physical and psychological reactions and social behavioural reactions.

No significant differences were observed in the score of the complete instrument according to the university of origin of the participants. Neither, in stressors, physical and psychological reactions, nor in social behavioural reactions (Table 3).

When evaluating the differences between careers of origin (Fig. 1), small differences were found for the complete instrument (eta2 = 0.021). Likewise, small differences were observed in the subscales of stressors, physical and psychological reactions, and coping. However, no significant differences were found for social behavioural reactions, with effect sizes close to 0.01, and very similar for all variables. Figure 1 shows that Dentistry has the highest values in stressors, physical and psychological reactions, as well as social behavioural reactions. On the other hand, Medical Technology has the lowest scores in stressors and physical and psychological reactions. As regarding Chemistry and Pharmacy, show the lowest scores in coping strategies.

**Table 1 Distribution of participants according to sex, university and career.**

| Career | UDEC (518) | | UDD (399) | | UCSC (132) | |
|---|---|---|---|---|---|---|
| | Men (126) | Women (392) | Men (105) | Women (294) | Men (29) | Women (103) |
| Nursing | 0 | 0 | 28 | 142 | 13 | 62 |
| Speech Therapy | 9 | 55 | 3 | 33 | 0 | 0 |
| Kinesiology | 21 | 33 | 30 | 15 | 0 | 0 |
| Nutrition and Dietetics | 4 | 75 | 7 | 25 | 0 | 0 |
| Obstetrics | 7 | 76 | 0 | 0 | 0 | 0 |
| Dentistry | 0 | 0 | 37 | 79 | 0 | 0 |
| Chemistry and Pharmacy | 45 | 67 | 0 | 0 | 0 | 0 |
| Medical Technology | 40 | 86 | 0 | 0 | 16 | 41 |
| Total | 126 (24.32) | 392 (75.68) | 105 (26.32) | 294 (73.68) | 29 (21.97) | 103 (78.03) |
| | 518 (49.38) | | 399 (38.04) | | 132 (12.58) | |

Notes.

UDEC, Universidad de Concepción; UDD, Universidad del Desarrollo; UCSC, Universidad Católica de la Santísima Concepción.

**Table 2 Comparison of SISCO-II-EA scores among participants by sex and effect size (Cohen's d).**

| Variables | Women | | Men | | t-statistic | p-value | d |
|---|---|---|---|---|---|---|---|
| | M | SD | M | SD | | | |
| Stressors | 3.298 | 0.595 | 3.010 | 0.671 | $t(399.6) = 6.16$ | **<0.001** | 0.468 |
| Physical and psychological reactions | 3.210 | 0.683 | 2.770 | 0.739 | $t(412.5) = 8.48$ | **<0.001** | 0.631 |
| Social behavioural reactions | 2.765 | 0.825 | 2.567 | 0.850 | $t(429.1) = 3.27$ | **=0.001** | 0.237 |
| Total reaction | 3.053 | 0.675 | 2.698 | 0.712 | $t(420.9) = 7.05$ | **<0.001** | 0.518 |
| Coping strategies | 3.140 | 0.632 | 3.114 | 0.673 | $t(417.6) = 0.55$ | 0.585 | 0.040 |
| Full instrument | 3.133 | 0.474 | 2.859 | 0.524 | $t(405.5) = 7.47$ | **<0.001** | 0.562 |

Notes.

M, arithmetic mean; SD, standard deviation; p-value, probability associated with the t-statistic; d, Cohen's d.
Bold represents significant differences ($p \leq 0.001$).

**Table 3 Comparison of SISCO-II-EA scores between university of origin, and effect size (Eta squared).**

| Variables | UdeC | | UDD | | UCSC | | F-statistical | p-value | eta² |
|---|---|---|---|---|---|---|---|---|---|
| | M | SD | M | SD | M | SD | | | |
| Stressors | 3.257 | 0.621 | 3.203 | 0.634 | 3.178 | 0.626 | $F(2, 1,044) = 1.31$ | 0.271 | 0.002 |
| Physical and psychological reactions | 3.075 | 0.699 | 3.130 | 0.749 | 3.115 | 0.736 | $F(2, 1,044) = 0.68$ | 0.505 | 0.001 |
| Social behavioural reactions | 2.723 | 0.808 | 2.690 | 0.853 | 2.765 | 0.894 | $F(2, 1,044) = 0.44$ | 0.645 | 0.001 |
| Total reaction | 2.951 | 0.677 | 2.975 | 0.720 | 2.992 | 0.737 | $F(2, 1,044) = 0.24$ | 0.787 | 0.000 |
| Coping strategies | 3.141 | 0.636 | 3.165 | 0.639 | 3.011 | 0.667 | $F(2, 1,044) = 2.92$ | 0.055 | 0.006 |
| Full instrument | 3.067 | 0.487 | 3.07 | 0.528 | 3.043 | 0.509 | $F(2, 1,044) = 0.15$ | 0.861 | 0.000 |

Notes.

UDEC, Universidad de Concepción; UDD, Universidad del Desarrollo; UCSC, Universidad Católica de la Santísima Concepción; M, arithmetic mean; SD, standard deviation; p-value, probability associated with $F$ statistic; eta², Eta squared.

Of the total number of participants, 6.96% (73) reported receiving psychological treatment, and of these, 37% (*Castillo-Navarrete et al., 2023b*) were taking medication. In the context of obtaining the SISCO-II-AS scale in college students, it was decided to include

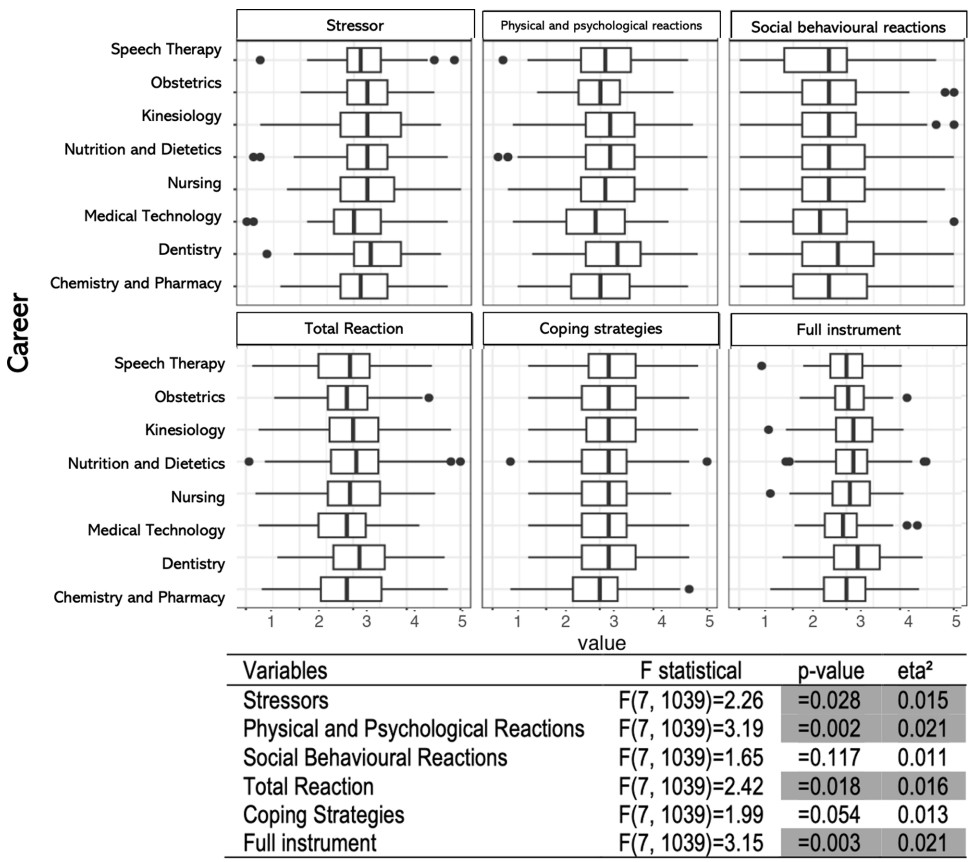

| Variables | F statistical | p-value | eta² |
|---|---|---|---|
| Stressors | F(7, 1039)=2.26 | =0.028 | 0.015 |
| Physical and Psychological Reactions | F(7, 1039)=3.19 | =0.002 | 0.021 |
| Social Behavioural Reactions | F(7, 1039)=1.65 | =0.117 | 0.011 |
| Total Reaction | F(7, 1039)=2.42 | =0.018 | 0.016 |
| Coping Strategies | F(7, 1039)=1.99 | =0.054 | 0.013 |
| Full instrument | F(7, 1039)=3.15 | =0.003 | 0.021 |

**Figure 1** **Differences between the careers by the university of origin.** Show differences founded between the careers according to the university of origin.

those receiving psychological and pharmacological treatment. These individuals are part of the student reality and can perform academically (*Fu & Qiao, 2023*; *Lieslehto et al., 2023*; *Tang et al., 2023*). This group of participants showed moderate differences in the score of the full instrument ($d = 0.614$). When analyzing the subscales individually, significant differences were found in all except coping (Table 4). Specifically, a small difference was observed in stressors ($d = 0.225$). The differences were greater in physical and psychological reactions ($d = 0.766$) and social behavioural reactions ($d = 0.641$).

When examining the relationship between the age of the participants and the instrument, a linear relationship close to $r = 0$ was mostly observed (Fig. 2). However, the coping subscale showed a significant relationship with age, although of a small magnitude ($r = 0.09$, $p < 0.01$).

Considering that differences by university and career are minimal, differences by age are insignificant, and differences in stress level are predictable in individuals on psychological and pharmacological therapy, sex is identified as the relevant variable in setting norms. There is no clear explanation for why a group should show significant differences in the subscales or the entire instrument. Therefore, norms are established for each subscale

**Table 4  Comparison of SISCO-II-AS scores between participants with and without psychological and pharmacological therapy, and effect size (Cohen's d).**

| Variables | With therapy | | Without therapy | | t-statistic | p-value | d |
|---|---|---|---|---|---|---|---|
| | M | SD | M | SD | | | |
| Stressors | 3.358 | 0.685 | 3.217 | 0.621 | t(79.2) = 1.71 | 0.092 | 0.225 |
| Physical and psychological reactions | 3.606 | 0.678 | 3.063 | 0.712 | t(82.4) = 6.58 | **<0.001** | 0.766 |
| Social behavioural reactions | 3.208 | 0.821 | 2.679 | 0.825 | t(81.3) = 5.30 | **<0.001** | 0.641 |
| Total reaction | 3.466 | 0.641 | 2.928 | 0.691 | t(83.1) = 6.88 | **<0.001** | 0.783 |
| Coping strategies | 3.002 | 0.643 | 3.144 | 0.641 | t(81.2) = 1.81 | 0.074 | 0.220 |
| Full instrument | 3.348 | 0.476 | 3.044 | 0.496 | t(82.2) = 5.25 | **<0.001** | 0.614 |

**Notes.**

M, arithmetic mean; SD, standard deviation; p-value, probability associated with the t-statistic; d, Cohen's d.

Bold represents significant differences ($p \leq 0.001$).

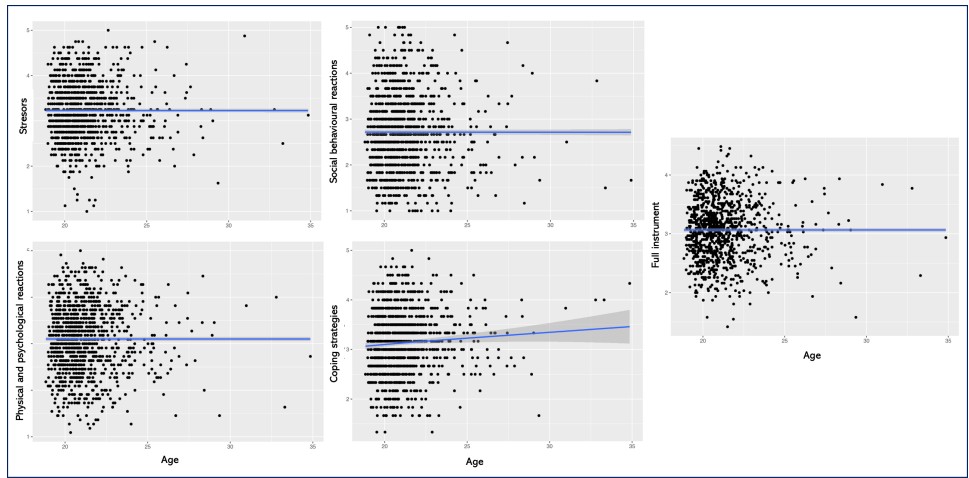

**Figure 2  Scatter plot showing the relationship between age and the SISCO-II-AS.** Scatter plot showing the relationship between age and the subscales of stressors, physical and psychological reactions, social behavioural reactions, total reaction, coping strategies and for the whole instrument.

and the full instrument using percentile <25th percentile, 25th–50th percentile and >50th percentile cut points for low, medium, and high levels, respectively (*Gutiérrez et al., 2016*). A summary of the norms established for the full instrument and its subscales is presented in Table 5. For a more detailed analysis, a breakdown by percentile and gender is available as supplementary material.

For the stressors subscale, it is considered low level if women score ≤ 23 and men ≤ 20. The mean score is between 24 and 29 for women, and between 21 and 27 for men. A high score corresponds to ≥ 30 for women and ≥ 28 for men. In the physical and psychological reactions subscale, a high score is reached with ≥ 41 for women and ≥ 36 for men. The intermediate score varies between 31 and 40 for women, and between 26 and 35 for men. Thus, ≤ 30 for women and ≤ 25 for men indicate a low level on this subscale. For social behavioural reactions, a high score is ≥ 20 for women and ≥ 19 for men. A medium score

**Table 5  Summary of the standards set for SISCO-II-AS (full instrument and its subscales).**

| Variable | SISCO-II-AS score | | | | | |
| --- | --- | --- | --- | --- | --- | --- |
| | Women | | | Men | | |
| | Low AS | Medium AS | High AS | Low AS | Medium AS | High AS |
| Stressors | $\leq 23$ | 24–29 | $\geq 30$ | $\leq 20$ | 21–27 | $\geq 28$ |
| Physical and psychological reactions | $\leq 30$ | 31–40 | $\geq 41$ | $\leq 25$ | 26–35 | $\geq 36$ |
| Social behavioural reactions | $\leq 12$ | 13–19 | $\geq 20$ | $\leq 11$ | 12–18 | $\geq 19$ |
| Total reaction | $\leq 44$ | 45–59 | $\geq 60$ | $\leq 38$ | 39–53 | $\geq 54$ |
| Coping strategies | $\leq 16$ | 17–21 | $\geq 22$ | $\leq 16$ | 17–21 | $\geq 22$ |
| Full instrument | $\leq 86$ | 87–107 | $\geq 108$ | $\leq 78$ | 79–98 | $\geq 99$ |

**Notes.**
Low AS,  Low level of AS given by centiles 0–25;  Medium AS,  Medium level of AS given by centiles 25–75;  High AS,  High level of AS given by centiles 75–100.

for women is in the range of 13 to 19 points, and for men, it is in the range of 12 to 18 points. Therefore, $\leq 12$ for women and $\leq 11$ for men indicates a low level in this subscale.

In the total score of the total reaction subscale, which includes physical and psychological reactions and social behavioural reactions (*Castillo-Navarrete et al., 2020*), a high level is considered when women score $\geq 60$ points and men score $\geq 54$ points. A medium level is between 45 and 59 points for women, and between 39 and 53 points for men. On the other hand, $\leq 44$ for women and $\leq 38$ for men will correspond to a low level. In coping strategies, a high score is established for women and men scoring $\geq 22$ points. A mild score is considered for those with scores between 17 and 21, while a low score corresponds to $\leq 16$.

When evaluating the full instrument, for women it is established that a high level of AS is reached with $\geq 108$ points (Table 5). A medium level is in the range of 87 to 107 points, while $\leq 86$ points corresponds to a low level. For men, a high level of AS is achieved with $\geq 99$ points. A medium level ranges from 79 to 98 points, and $\leq 78$ points indicates a low level.

Summarising the key findings, our analysis identified significant differences in several variables. Significant differences by sex were found for stressors, physical and physiological reactions, social behaviour, and the entire instrument, which reaffirms the importance of considering sex when studying academic stress. In addition, the scale of scores on the SISCO-II-AS and its sub-scales is essential for an accurate and contextual interpretation of academic stress levels, allowing for meaningful comparisons within the student population.

## DISCUSSION

AS is a common experience in the lives of college students, potentially leading to negative physical and mental health consequences. Adequate assessment of academic stress is crucial to identify students at risk and provide the necessary intervention. The present study aimed to descriptively analyse the SISCO-II-AS and to obtain its corresponding scales. The instrument was administered before the COVID-19 pandemic (face-to-face academic activities). During the pandemic, academic conditions underwent significant changes.
However, with the return to traditional academic activities, a proper objectification of academic stress is essential.

The SISCO-II-AS has strong psychometric properties (*Castillo-Navarrete et al., 2020*; *Guzmán-Castillo et al., 2022*; *Castillo-Navarrete et al., 2023b*). These ensure the reliability and validity of the instrument in measuring academic stress in college students. To interpret the scores obtained in the instrument, the respective bar norms are required. These set percentile cut-off points for categorising AS levels into low, medium, and high (Table 5 and Supplemental Material). This provides a clear guide for the interpretation of the scores. The instrument also takes into account gender differences in its scores. This reflects the well-documented differences in stress response between men and women (*Goldfarb, Seo & Sinha, 2019*; *Graves et al., 2021*; *Kuhn et al., 2023*; *Matud, 2004*). These gender differences in the stress response are observed at biological, psychological, and social levels (*Tamres, Janicki & Helgeson, 2002*).

The present study shows several important implications for the analysis of AS. First, it is notable that the gender distribution of the sample is skewed towards women, who represent 75.21% of the participants. Our sample, with a higher number of women, specifically reflects the demographics of accessible health faculties, not a global trend. The selection was based on the availability and cooperation of such faculties, which explains the absence of majors such as engineering. The detection of significant differences in the total scores of the SISCO-II-AS and its subscales according to the gender of the participants is consistent with existing literature, which suggests that women and men may experience and manage stress differently (*Matud, 2004*). Specifically, significant differences were found in stressors, physical and psychological reactions, and social behavioural reactions. The observed gender differences in SISCO-II-AS scores can be explained by several reasons. For example, on the stressors subscale, women have a low-stress score if they score 23 or less, while for men the threshold is 20. It is plausible to posit that women might be more susceptible to certain academic stressors or have a greater willingness to report their stressful experiences. These gender differences may be attributable to biological, psychological, and socio-cultural factors, and highlight the need to take gender into account in the assessment and management of AS.

The findings also reveal that, despite significant differences in physical and psychological reactions as well as in social behavioural reactions, no significant differences were observed on the coping strategies subscale according to gender. This result is in line with studies suggesting that gender differences in coping may be less pronounced in specific contexts, such as academia (*Tamres, Janicki & Helgeson, 2002*). However, one should not overlook what has already been reported regarding the weakness of the coping sub-scale (*Guzmán-Castillo et al., 2022*). Therefore, it is important not to forget that these interpretations require further research to fully understand the neuropsychophysiological causes underlying the stress response.

Regarding the university of origin, no significant differences were found in the scores of the whole instrument or the subscales. When disaggregated by degree, it stands out that dental students obtained the highest scores on the SISCO-II-AS. However, this does not differ from what is reported in the literature (*Avramova, 2023*; *Owczarek, Lion*

*& Radwan-Oczko, 2020*). Dental students face high levels of stress due to the rigorous demands of their academic and practical training. In addition, they are under pressure when dealing with patients and facing complex clinical situations from the early stages of their training. Fear of making mistakes and long hours of study and practice can also increase stress. In addition, dentistry involves a significant financial investment, which adds additional pressure and anxiety (*Avramova, 2023*; *Owczarek, Lion & Radwan-Oczko, 2020*).

In our study, medical technology students exhibited lower scores in stressors and physical and psychological reactions. This trend might be attributed to the holistic nature of their training, which not only emphasizes a solid grounding in ethical and scientific-technical aspects but also incorporates a significant practical and applied work component. This practical approach could help students connect their learning with real-life situations, potentially reducing anxiety and stress related to theoretical learning and exams. In contrast, students in chemistry and pharmacy displayed the lowest scores in coping strategies. This finding suggests that, while students across different majors may experience comparable levels of stress, the coping strategies they employ might differ. This is consistent with the literature indicating that coping is a dynamic process influenced by a variety of factors, including the educational environment (*Folkman & Moskowitz, 2004*).

The present study also reveals that students undergoing psychological and pharmacological treatment have higher levels of stress. This underlines the importance of considering mental health in the assessment and management of AS. This finding supports research demonstrating the high prevalence of mental health problems among college students (*Auerbach et al., 2018*).

It is crucial to acknowledge certain limitations of this study. Primarily, our sample is predominantly composed of students from specific universities and fields of study, which could potentially limit the generalizability of our results to broader student populations. The observed gender imbalance is reflective of the demographic makeup of the faculties and programs accessed, consistent with patterns noted in our previous research (*Castillo-Navarrete et al., 2023a*). This may affect the presentation and metrics of academic stress and associated biomarkers like BDNF and the percentage of global DNA methylation. We've attempted to mitigate these issues through gender-stratified analyses, but we recommend interpreting our findings with care, particularly when extending them to more gender-balanced groups. Secondly, the cross-sectional design of our study restricts our capacity to establish causality. While correlations between AS and various factors have been identified, the directional nature of these relationships remains unclear. Thirdly, the reliance on self-reported measures, such as the SISCO-II-AS inventory, introduces the possibility of response biases, including the tendency toward socially desirable answers, which might skew the results. Lastly, the study does not account for external variables that could significantly impact AS, such as familial, financial, or environmental factors, nor does it examine the nuances of individual learning experiences and workloads. The exclusion of these elements could constrain a more nuanced understanding of AS. Future research should aim to integrate these considerations to enhance our comprehension of the academic stress landscape.

This study exhibits crucial strengths in terms of its relevance and applicability. First, it employs a large and diverse sample, composed of students from different disciplines, enhancing the representativeness of the data. This broad representation reinforces the generalisability of the results to a large and varied student population. In addition, it provides a comprehensive and valuable insight into the prevalence and dynamics of academic stress among college students. In this regard, it identifies differentiating factors according to the field of study and gender. Furthermore, the differential impact of AS on men and women highlights the need for supportive approaches that take these gender differences into account. Although the study has limitations, its findings contribute significantly to the understanding of AS in the university context, providing useful clues for future research and guiding intervention strategies aimed at improving students' mental health and academic performance.

In conclusion, this study contributes to the field of AS research by providing a descriptive analysis of the SISCO-II-AS inventory and establishing norms for its interpretation. These norms are useful for health professionals, educators and other specialists working with college students, as they allow them to more accurately assess the level of AS and design appropriate interventions to promote student well-being.

## ACKNOWLEDGEMENTS

We would like to thank to our study participants for their involvement. Some sections of this paper were written with the help of the GPT-4 AI model. A document detailing the procedure used for the improvement of writing in English is available as supplementary material. However, the results of this study are presented clearly, honestly, and without fabrication, falsification, or inappropriate data manipulation.

### Funding
This work was supported by Grant 2021000311MUL of the Vice Rector of Research and Development (VRID) of the Universidad de Concepción and by Grant DIREG 01/2021 of the Directorate of Research of the Universidad Católica de la Santísima Concepción. Doctoral Scholarship in Chile ANID N°21201061 and N°21160620. The funders had no role in study design, data collection and analysis, decision to publish, or preparation of the manuscript.

### Grant Disclosures
The following grant information was disclosed by the authors:
Universidad de Concepción: 2021000311MUL.
Directorate of Research of the Universidad Católica de la Santísima Concepción: DIREG 01/2021.
Chile ANID: 21201061, 21160620.

## Competing Interests

The authors declare there are no competing interests.

## Author Contributions

- Juan-Luis Castillo-Navarrete conceived and designed the experiments, performed the experiments, analyzed the data, prepared figures and/or tables, authored or reviewed drafts of the article, and approved the final draft.
- Claudio Bustos conceived and designed the experiments, performed the experiments, analyzed the data, prepared figures and/or tables, authored or reviewed drafts of the article, and approved the final draft.
- Alejandra Guzman-Castillo performed the experiments, analyzed the data, authored or reviewed drafts of the article, and approved the final draft.
- Walter Zavala performed the experiments, analyzed the data, authored or reviewed drafts of the article, and approved the final draft.

## Data Availability

The data is available at Repositorio de Datos—UdeC: Castillo-Navarrete, Juan-Luis; Bustos, Claudio; Guzmán-Castillo; Alejandra; Zavala, Walter, 2023, "Academic Stress in College Students: Descriptive Analyses and Scoring of the SISCO-II Inventory", https://doi.org/10.48665/udec/M6681K, Repositorio de Datos—UdeC, V2.

## Supplemental Information

Supplemental information for this article can be found online at http://dx.doi.org/10.7717/peerj.16980#supplemental-information.

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
