# Peer review of "Academic stress in college students: descriptive analyses and scoring of the SISCO-II inventory"

_PeerJ, doi:10.7717/peerj.16980_

## Round 0.1 · original submission · Major Revisions

Dear Authors,

I would like to see some examples of academic stress and severe mental health conditions in the background of the research. Secondly, the justification for doing this study is not clearly explained.

Reviewer 1 ·

Basic reporting

This study uses a wide range of heterogeneous samples to make the data more representative. Research provides a descriptive analysis of the SISCO-II-AE inventory and estimates bar norms for assessing academic stress in college students. I commend the authors for their extensive data set, compiled over many years of detailed fieldwork. In addition, the manuscript is clearly written in professional, unambiguous language.
1. I recommend the authors to include a paragraph explaining the structure of the remaining of the manuscript, at the end of section "introduction".
2.From lines 86 to 88, the author can cite some relevant research. Here, I suggest quoting “Trajectories of college students' general self-efficacy, the related predictors, and depression: A piecewise growth mixture modeling approach. Heliyon, 9(5), e15750. https://doi.org/10.1016/j.heliyon.2023.e15750”,to supplement some studies on depression, gender, and other factors or pathways related to academic stress.

Experimental design

1.The most important question is whether there is no reasonable explanation for the high number of female participants in the sample, which will have a significant impact on the data analysis of the article. Is the high number of female participants related to the school and major chosen by the author? I think the author can increase this aspect of analysis. If there have been previous studies that have reflected similar issues, the author can introduce them in the introduction section.
2.The next most important issue is the lack of analysis on data preprocessing in Methodology, such as the author's lack of explanation on the handling of missing values.

Validity of the findings

1.Multiple evaluation tools can be listed in lines 92 to 93, rather than simply saying that there are many evaluation tools that make it complex. For complex arguments, some more examples can be provided to enrich the paragraph content. In addition, I believe that the content in lines 95-103 overlaps with the complexity mentioned in lines 91-93, and I believe the author can rethink how to make the logic between paragraphs clearer.
2.The descriptive analysis at the beginning of line 179 lacks capitalization.
3.The author can add another paragraph in the results section, summarizing which variables have significant differences and which have not been briefly summarized.

Additional comments

The limitations of the introduction section in lines 371 to 374 are that the author can select some content to cite relevant literature in the introduction section for introduction. Here, the reference I mentioned above can also be cited to enrich the research content.

·

Basic reporting

The article meets the journal’s guidelines. Ethical approval statements have been checked.
The data has been deidentified and the experiments conducted have been ethical.
The supplemental files, the figures and tables in the manuscript have been checked. The sample sizes have been confirmed as well.

Experimental design

no comment

Validity of the findings

Revisions
Table 3, column 4 is named “statistical”, maybe rename it to F-statistic.
Comment on the unbalanced gender categories (75% vs 25%) in the study sample.
Rephrase line 58-59, sentence starting with “Its finding…”.
Citing the second sentence in the introduction will strengthen the argument. Consider citing the sentence starting with “Consequently …” on line 82. Great references have been provided when discussing the SISCO system. Was there any missing data? If yes, how did the authors deal with it.
Line 307-310, the UNESCO does not suggest such heavy imbalance though. Please provide a few comments if the study sample was selected randomly or by any other procedure. Were the student majors selected at random? Why weren’t there any engineering students selected, for example?
Overall, the authors have demonstrated a strong framework to analyze categorical survey related data. However, a few arguments require more strengthening.

---

## Round 0.2 · accepted · Accept

Thanks for addressing the changes in the manuscript.

Reviewer 1 ·

Basic reporting

The author has made revisions and I have no further comments.

Experimental design

The author has made revisions and I have no further comments.

Validity of the findings

The author has made revisions and I have no further comments.

·

Basic reporting

Authors have addressed the comments in my previous review of the article.
The article needs to further examination and is ready to be published.

Experimental design

NA

Validity of the findings

NA

Additional comments

NA